# Effects of Cha-Cha Dance Training on the Balance Ability of the Healthy Elderly

**DOI:** 10.3390/ijerph192013535

**Published:** 2022-10-19

**Authors:** Han Li, Xuan Qiu, Zhitao Yang, Zhengxiao Zhang, Gang Wang, Youngsuk Kim, Sukwon Kim

**Affiliations:** 1Department of Physical Education, Jeonbuk National University, Jeonju 54896, Korea; 2Department of Physical Education, Huaiyin Normal University, Huaian 223000, China; 3Department of Physical Education, Yichun University, Yichun 336000, China; 4Department of Mechanical and Electrical Engineering, Zhoukou Normal University, Zhoukou 466000, China; 5Department of Physical Education, Xinyang Normal University, Xinyang 464000, China

**Keywords:** Cha-cha, elderly, balance, intervention

## Abstract

Neuro-musculo-skeletal degradations with advancing age are generally accompanied with mobility problems and poor health status, contributing to impaired physical function and increased risk of falls. In an effort to diminish a fall-related risk such as unstable balance, a variety of interventions have been studied and implemented. Yet, there have been few studies to evaluate the effect of Cha-cha dance training on postural balance or balance performance of the elderly. The Cha-cha dance is composed of moderate rhythm and symmetrical movements and is easy to master and even easier for the elderly to begin. The purpose of this study was to intervene the fitness exercise through 12-week Cha-cha dance training, evaluate its impact on the balance ability of the elderly, and provide a scientific experimental basis for the elderly to participate in the Cha-cha dance exercise. Forty healthy elderly people with no difference in balance ability were randomly divided into two groups. The Cha-cha training group (CTG, n = 20) regularly participated in Cha-cha dance class, 3 times a week, 90 min each time, for a total of 12 weeks, and the control group (CONG, n = 20) did not participate in the class and engaged to their regular daily life. Dynamic and static balance were measured at pre and post intervention. Overall, the results showed that dynamic balance and static balance in CTG were significantly improved after the intervention. In addition, the results showed that the improvement was more significant in trials in left foot than right foot, and trials in closed eyes than open eyes, respectively. In conclusion, a 12-week Cha-cha dance class or program alone can be an effective intervention to improve balance of the elderly.

## 1. Introduction

Falls are the second leading cause of unintentional injury death worldwide, and the largest number of unintentional deaths due to falls in adults over the age of 60 [1]. With advancing age, the physical functions of the elderly, such as vision and hearing ability, muscle strength, and reaction and coordination ability, constantly decline, and failure of detecting improper body posture or regaining loss of balance increase the likelihood of falls [2,3,4,5,6]. Body posture or balance cannot be maintained properly during daily activities without accurate integrations amongst sensory input, central processing, and muscle force development [6].

Lower limb muscle strength declines by 40% between the ages of 30 and 80 [5], and the risk of falls continues to increase with advancing age. After age 60, older adults have significantly reduced balance and increased risk factors for falls and injuries [6]. More than 30% of the elderly over the age of 65 fell at least once a year on average, and one-fifth of them required different degrees of medical treatment [7].

Dancing is a complex sensory rhythmic exercise that incorporates a variety of physical, cognitive, and social factors that may improve fall risk factors [8]. A dance-based study of older adults found that dancing was an effective fall prevention strategy [9] and that it was also effective in improving gait and balance [10]. A randomized controlled trial (RCT) of fifty participants also showed that different dance styles improved balance and gait speed in older adults [11,12,13,14,15].

Long-term Latin dance training helps to develop endurance, speed, flexibility, and many other physical qualities [7,16,17]. Not only that but Latin dance training helps to improve stability [7,17]. Latin dance had a positive effect on flexibility, balance, agility, dynamic balance ability, and knee flexion strength of the elderly, suggesting that dance sports training was very effective in preventing fall accidents in the elderly [18]. According to the World Sports Dance Federation (WDSF), adjudicators (a panel of judges) evaluate a dance performance in five components, and, among them, posture, balance, and coordination (PB) include body posture, body line, shape, body holding and posture design, body position, and change. Latin dancers are supposed to build strong and flexible ankles because the ankle joint is an important weight-bearing joint when transitioning and/or balancing body posture [7,16].

The proper function of ankle joint (i.e., ankle strategy) is important in means of postural balance [19]. The ankle joint stability and flexibility continue to deteriorate with advancing age and it becomes more difficult to maintain balance which contributes to an increase in the likelihood of falls [20]. Latin dance training can enhance ankle joint stability and flexibility [21]. A study found that a training program employed Cha-cha, Rumba, and Jive effectively improved older adults’ walking balance and standing balance [17]. However, the more complex training method using Latin dance in the previous study is slightly more complicated for the elderly with weak muscle strength and reduced nerve innervation, especially for some elderly people with poor exercise ability [22,23].

Many studies have been conducted to improve the balance ability of the elderly by using a dance program that combines various dances. However, there is still a lack of research on enhancing balance ability of the elderly through Cha-cha dance intervention. Further, the present study tries to verify whether the intervention of the Cha-cha dance alone is effective in improving the balance ability of the elderly. Therefore, if the elderly is provided with a basic dance program whose exercise volume, exercise intensity, and exercise rhythm are suitable for the elderly’s staged physical fitness, it would be more practical to improve balance ability of the elderly. The present study evaluated whether a 12-week Cha-cha dance training would help improving the dynamic and static balance of the elderly.

## 2. Method

### 2.1. Subjects

Forty participants from a local senior university participated in the program for 12 weeks (Table 1). They all signed informed consent forms approved by the University Ethics Committee (JBNU2022-04-008-001). The experimenters were configured through random block assignment by generating random numbers in Excel (Figure 1); a Cha-cha training group (CTG), and a control group (CONG). The participants’ exclusion criteria were as follows: (1) The age conforms to the definition of the elderly (60–65 years old). (2) There is no habit of participating in regular sports for a long time. (3) There has been no fall or fall in the past two years. (4) In the past year, there has been no structural damage to the limbs and a series of diseases that affect movement. (5) No cardiovascular and cerebrovascular complications or blood pressure problems in the past year. (6) Mental health, no mental illness. (7) Willing to sign the participation agreement (including the safety agreement). The screening content mainly included two aspects; FMS functional exercise test and balance ability test [24] were used to ensure physical fitness and safety, and the balance ability level, respectively. There was no significant difference in the basic data of FMS physical function and balance between the two groups (*p* = 0.61) (Table 1).

### 2.2. Intervention

In the present study, CTG met three times per week, and each class lasted for 90 min with a 10-min break in the middle. CTG started from beginner level and progressed to intermediate level (Table 2). Target heart rate was set at 50~60% of HR max, about 75~90 bpm during the entire period, not exceeding 70% of HR max (about 105 bpm). A Polar watch (Polar Electro Inc., Bethpage, NY, USA) was used to monitor the heart rate (Table 3).

### 2.3. Measurements

Balance Check 636 (Dr-Wolff, Arnsberg, Germany), and Good Balance (Metitur Oy, Jyvaskyla, Finland) were used to test dynamic balance ability, and static balance ability of the groups, respectively. A total of six movements were tested for static balance, namely eyes open and feet standing (EO), eyes closed and feet standing (EC), eyes open and left foot standing (LEO), eyes closed and left foot standing (LEC), eyes open and right foot standing (REO), and eyes closed and right foot standing (REC).

Balance Check 636 reported three indicators of dynamic balance ability of the subjects; Score, Rot.Speed Max, and Rot.Speed Φ. When the subject starts to enter the test program, the red ball in the center of the area displayed on the screen will be sensed along with the uneven pressure on the sole of the foot caused by the slight shaking of the subject’s center of gravity, resulting in the movement of the red ball and the shaking process. The recorded node is 50 milliseconds. If the ball stays in an area for more than the time of the node, it will be recorded once. Each time it is recorded from the inside to the outside corresponds to the corresponding score. From the center area to the area 4, it is 30 points, 5 points, 2 points, 1 point, and 0 points. When the plantar pressure distribution is more stable, the less shaking, the better the control of the ball on the screen, therefore, the final score will also be high. Rot.Speed Max is the maximum angular velocity of the center automatically generated by the system from start to finish of a subject standing on the device. Rot.Speed Φ is the average angular velocity of the center of gravity automatically generated by the system from beginning to end of a subject standing on the instrument. The higher the Score value, the lower the Rot.Speed Max and Rot.Speed Φ values, the better the dynamic balance ability.

Good Balance reported five indicators of static balance ability of the subjects: C90 Area, Trace Length, Std Velocity, Std X Deviation, and Std Y Deviation. The smaller the value, the better the static balance ability. The subject needs to stand on the apparatus and hold it for 10 s. C90 Area is the area where the sole of the foot touches the ground. Shaking will cause the envelope area to increase. Trace Length focuses more on the area and the distance of conversion of pressure, which has a relationship-like stability. The larger the indicator, the worse the stability and the poorer the ability to balance. Std Velocity is the speed of the center of gravity shaking. When the amplitude of the center of gravity shaking is small, the speed of the shaking is also low, which means it is more stable. Std X Deviation and Std Y Deviation mainly evaluate the shaking speed, but the displacement direction of the velocity is separately counted. You can see the shaking speed of each person in different directions. The smaller the shaking speed, the lower the speed and the more stable in that direction.

### 2.4. Statistical Methods

Statical analysis was performed using the Graph Pad Prism 8.0 statistical program. All variables were tested for normal distribution using Mauchly’s sphericity test, and the results suggested satisfactory. In order to examine the exercise effect of a 12-week Cha-cha dance intervention, a two-way repeated measured ANOVA was performed. Main effect and interaction effect was found statistically significant if the *p* ≤ 0.05.

## 3. Results

### 3.1. Dynamic Balance Test Results

The results of Score, Rot.Speed Max, and Rot.Speed Φ were significantly different after training Table 4. There was a significant interaction of group and time. The results indicated that all tests for dynamics balance improved in CTG, but, not in CONG after the intervention.

### 3.2. Satic Balance Test Results

Results of 12-week cha-cha training on static balance, standing with eyes open are shown in Table 5. In the state of standing with eyes open, statistical differences were not found in all five indicators after 12 weeks but there were statistical “Group” differences in Std X Deviation and Std Y Deviation. There was a significant interaction effect in Std X Deviation.

Results of 12-week Cha-cha training on the static balance, standing with eyes closed and feet, are shown in Table 6. In the state of standing with eyes closed, Std Y Deviation showed statistical differences after 12 weeks. Main effects of “Group” was found significant in Trace Length, Std X Deviation and Std Y Deviation. In addition, interaction effects were found significant in Trace Length, Std Velocity, Std X Deviation, and Std Y Deviation.

Results of 12-week Cha-cha training on the static balance, standing with eyes open and left foot, are shown in Table 7. In the state of standing with eyes open and left foot, C90 Area, Trace Length, and Std X Deviation showed statistical differences after 12 weeks. Main effects of “Group” was found significant in C90 Area, Trace Length, Std Velocity, and Std X Deviation. In addition, interaction effects were found significant in C90 Area, Trace Length, Std Velocity, Std Velocity, and Std X Deviation.

Results of 12-week Cha-cha training on the static balance, standing with eyes closed and left foot, are shown in Table 8. In the state of standing with eyes closed and left foot, C90 Area, Trace Length, and Std X Deviation showed statistical differences after 12 weeks. Main effects of “Group” was found significant in C90 Area, Trace Length, and Std Velocity. In addition, interaction effects were found significant in C90 Area, Trace Length, Std Velocity, Std Velocity, and Std X Deviation.

Results of 12-week Cha-cha training on the static balance, standing with eyes open and right foot, are shown in Table 9. In the state of standing with eyes open and right foot, C90 Area and Std Deviation showed statistical differences after 12 weeks. Main effects of “Group” was found significant in Trace Length and Std Velocity. In addition, interaction effects were found significant in Trace Length.

Results of 12-week Cha-cha training on the static balance, standing with eyes closed and right foot, are shown in Table 10. In the state of standing with eyes closed and right foot, Trace Length, and Std X Deviation showed statistical differences after 12 weeks. Main effects of “Group” was found significant in Trace Length and Std Y Velocity. In addition, interaction effects were found significant in C90 Area and Trace Length.

After 12 weeks of Cha-cha dance intervention, the three dynamic balance indexes in CTG were significantly improved, and the five static balance indexes in CTG were significantly improved compared to CONG. The most obvious indicators of improvement are the length of the center of gravity rocking trajectory and the speed of the center of gravity on the *x*-axis. The improvement effect of the left foot was better than that of the right foot, and the improvement effect of closed eyes was better than that of open eyes.

## 4. Discussion

The present study evaluated if a 12-week Cha-cha dance would help improve the dynamic and static balance of the elderly. The results from the present study suggested that the intervention significantly improved both the dynamic and static balances of older adults.

In order to maintain proper balance when standing quietly, the synergy of nerves and muscles in the lower body is required [25,26,27,28]. As intermittent exercisers grow older, they show a decrease muscle strength [5], a decreased sensory function [29,30,31], and a decreased neuromuscular recruiting capacity and conduction [32,33], resulting in decreased ability to maintain postural stability [32,33,34,35,36]. Cha-cha dance contains the characteristics of many symmetrical movements, wide range of muscle mobilization, many steps per unit time, large moving distance, and fast transfer of center of gravity [37]. In the present study, these characteristics in Cha-cha, after the period of training, should have a positive effect on improving neuromuscular capacity of the elderly, enhancing their perception ability and strengthening the ability of nerves to mobilize muscles.

The three dependent variables (Score, Rot.Speed Max, and Rot.Speed Φ) in dynamic balance in CTG in comparison with that in CONG in the present study showed significant improvements after the training intervention. The training content for CTG mainly consisted of basic steps of performing repetitive symmetrical movements to improve the control ability of the small muscle groups and joints of the elderly. In addition, the strength of the abdominal and back extensors is required to maintain upper body stability when performing Cha-cha training [21]. The regular twisting of the upper body and the increase and release of pressure on the lower limbs during the Cha-cha dance process should help to improve the muscle support of the waist, abdomen, and lower limbs [21]. The stimulation of lower extremity muscles by repetitive basic movement training also strengthens the stability of ankles and knees in the elderly, and contributes to the development of joint flexibility [38]. Postural control during Cha-cha is the result of the interaction between the central nervous system and various tissues [39,40], and training-induced improvements in neural complementation responses may have enhanced dynamic and static balance of the elderly in the present study [41,42]. The biomechanical characteristics and movement characteristics of Cha-cha dance may have strengthened the proprioception; the repetitions of fast and slow movement exercises may have strengthened the inhibition of the differentiation of the nervous system in the present study [43,44]. The increased participation of nerves can be a good explanation for the enhancement of muscle strength [43]. Previous studies have shown that muscle strength, agility, balance, and flexibility were positively related to dance performance [44]. The improvement of the dynamic balance ability in the present study may be based on the improvement of the nerve’s ability to control the limbs, which in turn contributes to the gradual improvement of the body’s control ability and balance ability. Through 12 weeks of Cha-cha dance training, it was speculated that more effective muscle recruitment and mobilization ensured the quality of movement completion, thereby improving dynamic balance.

The five dependent variables (C90 Area, Trace Length, Std Velocity, Std X Deviation, and Std Y Deviation) in static balance in CTG in comparisons with that in CONG in the present study showed significant improvements after the training intervention. The basic steps of the Cha-cha dance are composed of mostly repetitive switching movements between the power leg and support leg. When feet are moving on the floor, they are supposed to cooperate with other joints or segments of the whole body to complete the action. At this interval, the rapid activation of the muscles of the lower limbs is required, and the stable posture of the upper body, when moving from one point to another point, must be maintained [45]. While the upper and lower limbs cooperate with each other, the feet play an important role in continuously moving the center of gravity of a dancer. This repetitive movement would help the control of the center of gravity and improve the ability to balance when moving or standing.

Response directional balance ability variables (Std X Deviation, Std Y Deviation) showed significant differences after 12 weeks of training intervention in the present study. This may be due to the characteristics of the male and female pair Cha-cha dance. Repeated practice of shifting the direction in forward-backward or in side-shifting while practicing the dance technique may have optimized leg muscle strength and neural control of the elderly in CTG [46]. In addition, a large difference in Std X Deviation in the present study seems to be influenced by repetitive practice of the lateral movement in the basic steps of Cha-cha dance.

Of the left foot and the right foot, the change of the dependent variable in the standing state of the left foot is more obvious. After the training intervention, the balance ability of the left foot in CTG was significantly improved regardless of whether the eyes were open or closed. The swing speed of the center of gravity in both axes of CTG was significantly better than that of CONG. In most people, the right leg is usually the dominant leg, and in these cases, the right leg is biased when standing, supporting weight, and performing coordinated movements. Over time, the left side becomes relatively weak, and the brain continues to reinforce the pattern of dominance on the right side [47]. Most of the movements of the Cha-cha dance are symmetrical movements. Under the condition that the left and right feet bear the same training load and intensity, and most people’s left limbs are not dominant, the lifting of the left foot is better than that of the right foot. The results of this experimental study have also verified the above pattern.

In the comparison between eyes open and eyes closed, the improvement of the static balance index of standing with eyes closed was more obvious, especially when standing with eyes closed on one foot, CTG showed better results because the visual perception ability was very important to the body posture and balance in the static state [48]. Then, the improvement of static balance ability in the closed eye state in the study also proves that the compensation effect of other sensory organs and muscle strength under poor vision is improved. This finding is also consistent with the findings of Rahal MA and Kattenstroth et al. [49,50]. When they were training the elderly in Tai Chi and physical dance, the dance group showed better performance in posture control when standing on one foot with eyes closed. Single-leg support balance is an important indicator in the evaluation of static balance ability, which is also a more common form in the Cha-cha dance movement system. It is precisely because there are many single-leg support movements in Cha-cha dance that it has a significant effect on improving static balance.

## 5. Conclusions

12-week Cha-cha training intervention significantly improved dynamic and static balance in older adults. Especially for the elderly do not often engage in physical exercise, a period of Cha-cha dance training can be used as an appropriate exercise to improve the executive ability of motor functions. In this study, no mid-term test was conducted during the experimental process, and there was no further research on the effect of Cha-cha dance training on the balance ability of the elderly in terms of timeliness and dose-effect. The choice of elderly age is limited, and there is little research on dance and postural balance. Therefore, the experiment will carry out follow-up in-depth research on the intervention of the elderly postural balance by the Cha-cha dance to explore its specific mechanism of action.

## Figures and Tables

**Figure 1 ijerph-19-13535-f001:**
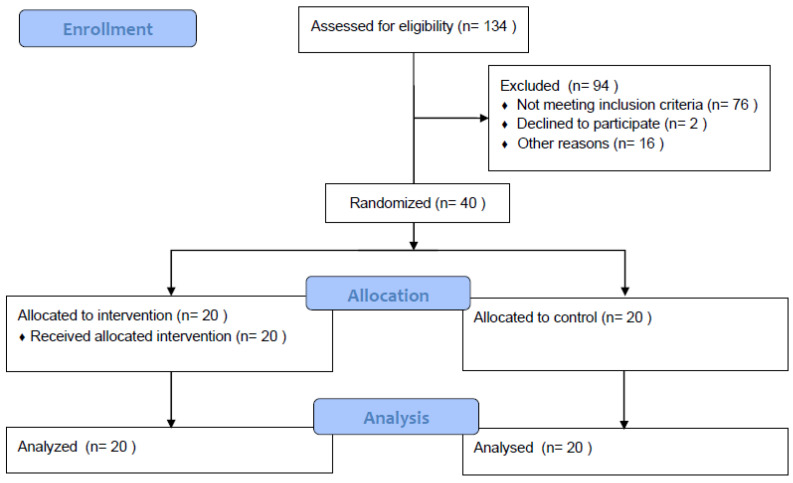
Study flowchart.

**Table 1 ijerph-19-13535-t001:** Basic information of experimental subjects.

Group	Age	Height (cm)	Weight (kg)	BMI (kg/m^2^)	FMS (Score)
CTG (N = 20)	61.75 ± 1.11	160.25 ± 6.25	63.40 ± 6.83	25.08 ± 2.62	10.45 ± 1.79
CONG (N = 20)	61.80 ± 1.23	160.20 ± 8.83	59.54 ± 5.95	24.36 ± 3.80	10.15 ± 1.90
*p*-value	0.89	0.98	0.06	0.49	0.61

**Table 2 ijerph-19-13535-t002:** Detailed Action Training Schedule.

Time	First Class	Second Class	Third Class
Week 1	Basic steps learning: (1) Basic movement (2) Forward Lock (3) Back Lock	1. Review of basic steps 2. Basic steps learning: (1) Hockey stick (2) New York (3) Underarm turn to Left	1. Review of basic steps 2. Basic steps learning: (1) Time step (2) Hand to Hand (3) Natural top
Week 2	1. Review of Basic steps 2. Basic steps learning: (1) Basic movement with 1/4 Turn (2) 1–5 Close Basic movement 3. Spot Turn	1. Review of basic steps 2. Basic steps learning: (1) Open hip twist (2) Natural Top	1. Review of basic steps 2. Basic steps learning: (1) Cuba rocks (2) Syncopate New York
Week 3–6	Single basic step combination exercise	Music rhythm single basic step combination practice	Male and female double exercise
Week 7–12	Music rhythm male and female double exercise	Male and female double ready for competition stage exercise	Male and female doubles competition stage

**Table 3 ijerph-19-13535-t003:** Exercise Load Control in CTG (heart rate).

Stage	Content	Period	HR (Times/Min)
Basic stage	Learn action	Weeks 1–2	68.48 ± 13.79
Reinforcement stage	Consolidation action	Weeks 3–6	78.78 ± 15.79
Improve stage	Improve action	Weeks 7–12	84.42 ± 17.54

**Table 4 ijerph-19-13535-t004:** Comparative analysis of various indicators of dynamic balance ability before and after the experiment.

Variables	Groups	Pre-Test	Post-Test	Time	Group	Interaction
Score	CTG	12,907.24 ± 739.88	16,345.75 ± 1835.32	≤0.0001 ***	≤0.0001 ***	≤0.0001 ***
	CONG	12,784.46 ± 812.85	12,698.36 ± 1461.65			
Rot.Speed Max (rad/s)	CTG	9.13 ± 1.96	4.39 ± 1.73	0.0003 ***	≤0.0001 ***	≤0.0001 ***
	CONG	9.31 ± 1.98	9.78 ± 1.56			
Rot.Speed Φ (rad/s)	CTG	3.69 ± 0.71	2.98 ± 0.59	0.0448 *	≤0.0001 ***	0.0024 **
	CONG	3.88 ± 0.63	4.22 ± 0.62			

Table note: CTG (n = 20), CONG (n = 20), * means *p* ≤ 0.05, ** means *p* ≤ 0.01, *** means *p* ≤ 0.001.

**Table 5 ijerph-19-13535-t005:** Comparative analysis of various indicators of static balance before and after the experiment with eyes open and feet standing (EO).

Variables	Groups	Pre-Test	Post-Test	Time	Group	Interaction
C90 Area (mm^2^)	CTG	585.34 ± 251.62	418.44 ± 187.59	0.2404	0.2187	0.0349 *
	CONG	550.6 ± 239.6	581.1 ± 161.4			
Trace Length (mm)	CTG	640.4 ± 199.3	485.1 ± 234.1	0.0993	0.1737	0.0838
	CONG	628.1 ± 173.3	632.9 ± 185.8			
Std Velocity (mm/s)	CTG	35.91 ± 9.553	29.5 ± 9.033	0.2419	0.3739	0.0633
	CONG	33.30 ± 6.877	35.13 ± 8.136			
Std X Deviation (mm/s)	CTG	9.500 ± 2.456	7.827 ± 0.907	0.6687	≤0.0001 ***	0.0045 **
	CONG	10.68 ± 1.653	11.99 ± 2.031			
Std Y Deviation (mm/s)	CTG	11.51 ± 2.138	10.50 ± 1.920	0.2022	0.0002 ***	0.1857
	CONG	12.76 ± 1.631	12.76 ± 2.097			

Table note: CTG (n = 20), CONG (n = 20), * means *p* ≤ 0.05, ** means *p* ≤ 0.01, *** means *p* ≤ 0.001.

**Table 6 ijerph-19-13535-t006:** Comparative analysis of various indicators of static balance ability before and after the experiment with eyes closed and feet standing (EC).

Variables	Groups	Pre-Test	Post-Test	Time	Group	Interaction
C90 Area (mm^2^)	CTG	873.9 ± 301.5	671.6 ± 260.0	0.3840	0.0809	0.0561
	CONG	885.8 ± 293.1	967.6 ± 391.9			
Trace Length (mm)	CTG	815.8 ± 238.0	533.8 ± 277.2	0.0501	0.0006 ***	0.0208 *
	CONG	843.1 ± 282.6	866.7 ± 139.0			
Std Velocity (mm/s)	CTG	61.89 ± 12.69	53.03 ± 9.724	0.2347	0.0738	0.0455 *
	CONG	60.31 ± 9.563	63.06 ± 7.996			
Std X Deviation (mm/s)	CTG	12.08 ± 3.225	8.650 ± 2.338	0.1425	≤0.0001 ***	≤0.0001 ***
	CONG	12.53 ± 3.113	14.05 ± 0.8823			
Std Y Deviation (mm/s)	CTG	14.66 ± 1.474	10.90 ± 1.176	≤0.0001 ***	0.0002 ***	≤0.0001 ***
	CONG	13.96 ± 1.509	14.38 ± 1.369			

Table note: CTG (n = 20), CONG (n = 20), * means *p* ≤ 0.05, *** means *p* ≤ 0.001.

**Table 7 ijerph-19-13535-t007:** Comparative analysis of various indicators of static balance ability before and after the experiment with eyes open and left foot standing (LEO).

Variables	Groups	Pre-Test	Post-Test	Time	Group	Interaction
C90 Area (mm^2^)	CTG	378,059 ± 8033	343,699 ± 5954	≤0.0001 ***	≤0.0001 ***	≤0.0001 ***
	CONG	377,730 ± 9671	375,492 ± 6663			
Trace Length (mm)	CTG	215,853 ± 743.8	172,885 ± 644.3	≤0.0001 ***	≤0.0001 ***	≤0.0001 ***
	CONG	215,845 ± 589.1	216,146 ± 519.5			
Std Velocity (mm/s)	CTG	654.6 ± 226.7	466.6 ± 163.3	0.5055	0.0033 **	0.0031 **
	CONG	645.8 ± 234.2	757.9 ± 228.3			
Std X Deviation (mm/s)	CTG	33.28 ± 8.470	22.57 ± 7.444	0.0357 *	0.0023 ***	0.0005 ***
	CONG	31.35 ± 7.063	34.87 ± 8.276			
Std Y Deviation (mm/s)	CTG	35.56 ± 7.709	28.04 ± 10.41	0.0686	0.1454	0.0569
	CONG	34.55 ± 7.112	35.52 ± 8.143			

Table note: CTG (n = 20), CONG (n = 20), * means *p* ≤ 0.05, ** means *p* ≤ 0.01, *** means *p* ≤ 0.001.

**Table 8 ijerph-19-13535-t008:** Comparative analysis of various indicators of static balance before and after the experiment with eyes closed and left foot standing (LEC).

Variables	Groups	Pre-Test	Post-Test	Time	Group	Interaction
C90 Area (mm^2^)	CTG	863,683 ± 4866	806,097 ± 8379	≤0.0001 ***	≤0.0001 ***	≤0.0001 ***
	CONG	865,058 ± 6327	867,001 ± 7952			
Trace Length (mm)	CTG	259,509 ± 703.2	221,689 ± 534.5	≤0.0001 ***	≤0.0001 ***	≤0.0001 ***
	CONG	259,344 ± 615.0	258,744 ± 1589			
Std Velocity (mm/s)	CTG	1587 ± 388.1	1214 ± 381.2	0.1390	0.0257 *	0.0086 **
	CONG	1489 ± 382.7	1608 ± 361.3			
Std X Deviation (mm/s)	CTG	59.33 ± 6.788	49.38 ± 5.886	0.0049 **	0.0823	0.0163 *
	CONG	57.86 ± 7.252	56.56 ± 8.038			
Std Y Deviation (mm/s)	CTG	56.98 ± 9.177	51.09 ± 7.465	0.0579	0.1790	0.1499
	CONG	57.02 ± 8.955	56.57 ± 7.658			

Table note: CTG (n = 20), CONG (n = 20), * means *p* ≤ 0.05, ** means *p* ≤ 0.01, *** means *p* ≤ 0.001.

**Table 9 ijerph-19-13535-t009:** Comparative analysis of various indicators of static balance before and after the experiment with eyes open and right foot standing (REO).

Variables	Groups	Pre-Test	Post-Test	Time	Group	Interaction
C90 Area (mm^2^)	CTG	912.2 ± 362.4	690.6 ± 165.0	0.0218 *	0.3220	0.1323
	CONG	888.1 ± 304.2	876.8 ± 295.1			
Trace Length (mm)	CTG	849.0 ± 245.7	598.6 ± 280.2	0.8388	0.0180 *	0.0191 *
	CONG	816.7 ± 286.1	1040 ± 492.5			
Std Velocity (mm/s)	CTG	212.6 ± 79.72	148.2 ± 72.07	0.0100 *	0.0346 *	0.3247
	CONG	234.2 ± 69.08	202.6 ± 63.94			
Std X Deviation (mm/s)	CTG	26.12 ± 8.561	18.24 ± 6.320	0.1071	0.1330	0.0563
	CONG	24.95 ± 7.946	26.01 ± 12.77			
Std Y Deviation (mm/s)	CTG	30.09 ± 7.707	24.71 ± 8.830	0.1904	0.1455	0.1004
	CONG	30.06 ± 9.135	31.49 ± 8.701			

Table note: CTG (n = 20), CONG (n = 20), * means *p* ≤ 0.05.

**Table 10 ijerph-19-13535-t010:** Comparative analysis of various indicators of static balance before and after the experiment with eyes closed and right foot standing (REC).

Variables	Groups	Pre-Test	Post-Test	Time	Group	Interaction
C90 Area (mm^2^)	CTG	1816 ± 452.7	1526 ± 402.4	0.2178	0.6578	0.0390 *
	CONG	1683 ± 578.1	1786 ± 358.9			
Trace Length (mm)	CTG	147,211 ± 245.7	112,556 ± 357.3	≤0.0001 ***	≤0.0001 ***	≤0.0001 ***
	CONG	147,102 ± 483.4	147,236 ± 363.6			
Std Velocity (mm/s)	CTG	462.5 ± 194.1	339.9 ± 172.8	0.1294	0.1011	0.0768
	CONG	454.6 ± 169.2	484.5 ± 142.4			
Std X Deviation (mm/s)	CTG	44.84 ± 9.201	36.66 ± 8.220	0.0365 *	0.1812	0.0549
	CONG	42.65 ± 7.662	42.78 ± 6.474			
Std Y Deviation (mm/s)	CTG	45.91 ± 9.070	39.86 ± 8.682	0.6058	0.0052 **	0.0603
	CONG	47.43 ± 10.51	50.83 ± 10.85			

Table note: CTG (n = 20), CONG (n = 20), * means *p* ≤ 0.05, ** means *p* ≤ 0.01, *** means *p* ≤ 0.001.

## Data Availability

The data presented in this study are available on request from the corresponding author.

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
