# Peer review of "Effects of Cha-Cha Dance Training on the Balance Ability of the Healthy Elderly"

_ijerph, 2022, doi:10.3390/ijerph192013535_

Round 1

Reviewer 1 Report

1. The second paragraph is accompanied by the loss of muscle strength elderly and Latin dance. Please separate the paragraph because it's a different content.

2. The limitations of Latin dance training are well explained. But the reason is that it is not justified to select Chacha dance as a research topic. Please add a references why you chose Chacha dance as the subject of your study.

3. In Table 1, height units are capitalized and weight is lowercase.  Please unify.

4. The period after the sentence is either after the table or before the sentence. Please edit it.

5. Please define C90, and Rot.Speed Φ. Readers may not understand meaning of the variables.

6. Please describe in detail the movement performed by the subject during the measurement.

7. I don't know exactly about the task motion, but I think the result of c90 is too high. I think you've probably integrated it, but I am not sure. Please check and describe how did you calculate it.

Reviewer 2 Report

Reviewers' comments to Authors:

The study addresses an important issue that valuated whether a 12-week Cha-cha dance training would help improving the dynamic and static balance of the elderly. This is very interesting study but quality of manuscrip should be extremely improved. The sample size is very small.

 The Authors should consider the following comments:

  • Abstract: The sentence 40 healthy elderly people, it should be Forty healthy elderly

o   Authors wrote that: Dynamic and static balance were measured at pre and post intervention. Please clarify the assessment of static and dynamic and which type of balance assessment did You used (e.g static or dynamic platforms or questionnaires). Add some p-values in abstract (Results)

·       Introduction. In abstract authors state that “Yet, there have been few studies to evaluate the effect of Cha-cha dance training on postural balance or balance performance of the elderly). Please add the references in introduction and discuss the main results. However, in introduction authors state” And so far, there is still a lack of research on enhancing balance ability of the elderly through Cha-cha dance intervention) – please clarify it. Please explain rationale of the study.

  • Methods paragraph is not adequately described. In Methods Consort for randomized studies diagram should be added. Please describe randomisation process. Please add more information about inclusion/exclusion criteria in study protocol (Page 4)

Table 1 should be below the text. BMI is expressed by %? Please check it.

FMS functional exercise test should be described

There was no significant difference in the basic data of FMS physical function and balance between the two groups (P>0.5). (Table 1). Please add exact values of p values.

PLease change treatment section for intervention section

All abreviations should be explained

Please explain all measured vriables  e.g, Score, Rot.Speed Max, Rot.Speed

  • Data in supplementary material are not in English
  • Statistical results should be better described: i.e which test was used to asses normality, did authors used post hoc tests?
  • Study has some limitations: small sample, lack of randomisation

Round 2

Reviewer 2 Report

Thank You for revised version of manuscript. Authors improved manuscript but they didn't add in relevant informations as suggested:

i.e desription randomisation process, Consort diagram, inclusion/exclusion criteria of the study.

Author Response

Comments: Authors improved manuscript but they didn't add in relevant information as suggested:

i.e desription randomisation process, Consort diagram, inclusion/exclusion criteria of the study.

Responses:

  • Randomization process, Consort plots, and study inclusion/exclusion criteria have been added to the first paragraph of the methods section.